# Association of Glycosylation-Related Genes with Different Patterns of Immune Profiles and Prognosis in Cervical Cancer

**DOI:** 10.3390/jpm13030529

**Published:** 2023-03-15

**Authors:** Wanling Jing, Runjie Zhang, Xinyi Chen, Xuemei Zhang, Jin Qiu

**Affiliations:** 1Department of Pharmacology, School of Pharmacy, Fudan University, Shanghai 200433, Chinazrj@shsmu.edu.cn (R.Z.); 2Obstetrics and Gynecology Department, Tongren Hospital, Shanghai Jiao Tong University School of Medicine, No.1111, XianXia Road, Shanghai 200336, China

**Keywords:** cervical cancer, glycosylation-related genes, immune microenvironment, prognosis, prediction models

## Abstract

(1) Background: Although the application of modern diagnostic tests and vaccination against human papillomavirus has markedly reduced the incidence and mortality of early cervical cancer, advanced cervical cancer still has a high death rate worldwide. Glycosylation is closely associated with tumor invasion, metabolism, and the immune response. This study explored the relationship among glycosylation-related genes, the immune microenvironment, and the prognosis of cervical cancer. (2) Methods and results: Clinical information and glycosylation-related genes of cervical cancer patients were downloaded from the TCGA database and the Molecular Signatures Database. Patients in the training cohort were split into two subgroups using consensus clustering. A better prognosis was observed to be associated with a high immune score, level, and status using ESTIMATE, CIBERSORT, and ssGSEA analyses. The differentially expressed genes were revealed to be enriched in proteoglycans in cancer and the cytokine–cytokine receptor interaction, as well as in the PI3K/AKT and the Hippo signaling pathways according to functional analyses, including GO, KEGG, and PPI. The prognostic risk model generated using the univariate Cox regression analysis, LASSO algorithm and multivariate Cox regression analyses, and prognostic nomogram successfully predicted the survival and prognosis of cervical cancer patients. (3) Conclusions: Glycosylation-related genes are correlated with the immune microenvironment of cervical cancer and show promising clinical prediction value.

## 1. Introduction

Globally, cervical cancer (CC) is one of the most frequent cancers amongst women [1]. Although advancements in screening techniques and treatment measures have greatly reduced the CC incidence and mortality in developed nations, CC remains a severe problem in developing countries [2]. With the continuous improvement in therapeutic measures, early CC is halted by surgery, achieving satisfactory results. However, metastatic CC is difficult to cure and often displays a poor prognosis owing to individual variability among patients. Recurrence and metastasis are pivotal hurdles in determining the survival and prognosis of CC, and there is no effective molecular marker to predict the prognosis. Thus, it is crucial to identify new biomarkers for the diagnosis and prognosis of CC.

The tumor immune microenvironment (TIME), including tumor cells, immune cells, and cytokines [3], is now recognized as a key factor in the development of cancer and chemo-resistance and has a significant impact on the expression of genes in cancer tissues [4]. Tumor and immune cells interact in the TIME as an essential process, shaping the progression of various cancer types, including CC [5]. For the precise therapeutic improvement of CC, it is vitally necessary to have a thorough understanding of the connection between the TIME and prognosis and to investigate innovative treatment approaches.

Studies have reported that some miRNAs and lncRNAs play a role in predicting CC. One previous study constructed a two-miRNA risk score model with predictive potential, providing new clues for the evaluation and treatment of CC [6]. In CC tissues and cell lines, miR-99a-5p expression was reported to be downregulated [7]. A study innovatively identified and validated four immune-related lncRNA signatures as predictors of CC [8]. In contrast to normal tissues, CC had significantly higher levels of EphA7 promoter methylation. EphA7 hypermethylation is therefore a promising signature to detect and screen CC [9]. Although these biomarkers can predict the survival and prognosis of patients with CC to some extent, they do not completely solve the problem.

Glycosylation is an important aspect among various post-translational modifications of proteins [10]. Glycosyltransferases and glycosidases regulate the majority of protein glycosylation in eukaryotes through the secretory pathway. A glycosidic link is created when the carbohydrates are transported to the protein’s amino acid residue [11]. Changes in glycosylation have been implicated to be intimately correlated with tumor cell invasion, metabolism, and immunity. A recent study reported that shortened O-Glycans could increase proliferation, impede differentiation, and cause invasive behavior by impairing cell–cell adhesion in adenocarcinomas [12].

Imbalanced glycosylation can influence the immune system in recognizing tumor cells and can modify glycan-binding receptors to induce an immunosuppressive response [13]. One of the primary characteristics of tumor cells is the glycosylation of the glycoproteins and glycolipids found on the cell surface. Tumor cells express glycosylation differently from the way in which normal cells do. As a result of the large number of different types of glycosylation-dependent lectin receptors expressed by immune cells, these cells are able to detect changes in glycosylation in the environment, which may induce immunosuppression [13]. Tumor cells can also camouflage themselves by expressing host-derived glycosylation and affect the expression of antigen-presenting cells, M2 macrophages, T cells, and NK cells, thereby promoting immune escape [14]. Carbohydrate Lewis antigens can attach to carcinoembryonic antigens expressed in colon cancer cells and combine with C-type lectin expressed in macrophages and immature dendritic cells to induce innate immune suppression [15,16]. The aggregation of Treg cells, the minimal infiltration of effector T cells, and the activity of NK cells are all strongly correlated with the sialylated structure of melanoma cells and the progression of the tumor in vivo [17]. The glycosylation of tumor cells usually occurs in the early stage of tumor development. In the prophase lesions of different types of tumors, some tumor-related glycosylation expression has been observed [18]. Therefore, the importance of glycosylation in cancer warrants further investigations to unmask the novel aspects of this hallmark.

In CC, the use of virus-induced glycosylated peptides for vaccines was originally reported more than four decades ago [19]. Recent research has discovered how glycosylation contributes to the development of CC and underscores the prospects of viable methods in distinguishing individual differences [20]. O-linked GlcNAcylation is used to influence major metabolic pathways [21]. Interestingly, elevated O-GlcNAcylation in CC was linked to increased cell proliferation and decreased cellular aging. Therefore, reducing O-GlcNAcylation could prevent the phenotypic transformation of HPV-18-transformed HeLa CC cells after treatment with appropriate inhibitors [22]. Glycans play key roles in the pathological processes of tumorigenesis and advancement. There is reduced expression of fucosylation in CC cytoplasmic proteins compared to normal tissues [23]. During carcinogenesis, dysregulated glycosyltransferases synthesize aberrant glycosylation structures, supporting tumor progression. Previous studies have demonstrated that differentially expressed genes (DEGs) of glycosyltransferase can predict the overall survival (OS) of pancreatic ductal adenocarcinoma patients and can be identified as prognostic markers [24]. Other studies have revealed that the expression of genes involved in glycosylation is very different in breast cancer compared to normal breast tissue [25]. Glycosylation-related genes have exhibited large expression variations between breast cancer subtypes, which may be associated with patient prognosis. However, the role of glycosylation-related DEGs in CC has remained poorly understood.

In this study, to investigate whether glycosylation-related genes are associated with differences in the TIME and prognosis of patients with CC, we thoroughly analyzed glycosylation-related DEGs in CC. A signature was developed to assess the prognostic value of glycosylation-related genes in CC. Our work is anticipated to offer new insights into the targeted therapeutic approach for CC.

## 2. Methods

### 2.1. Datasets and Samples

Glycosylation-related genes and CC samples were acquired from the Gene Set Enrichment Analysis (GSEA) Molecular Signatures Database and TCGA-GDC database. The following were the inclusion requirements: (a) samples with a CC diagnosis; (b) samples with a gene expression matrix and mapped clinical data; and (c) samples with all relevant clinical data, including age, FIGO stage, risk, and histopathological grade (Table 1). Samples without follow-up information were disqualified. Patients obtained from the TCGA-GDC database were randomly classified into training and testing cohorts for identification and validation.

### 2.2. Identification of Molecular Subgroups

According to the expression matrix of glycosylation-related genes, consensus clustering was carried out using the R program “Consensus Cluster Plus” to divide patients into two clusters [26]. Survival analysis between the two subgroups was also performed to assess the correlations among their survival rates.

### 2.3. Immune Analyses

Kyoto Encyclopedia of Genes and Genomes (KEGG) analysis illustrates major signaling pathways [27]. The Estimation of STromal and Immune cells in MAlignant Tumor tissues using Expression data (ESTIMATE) algorithm was used to determine the stromal score, immune score, and estimation score [28]. To quantify the relative proportions of different types of immune cells in the tumor sample, CIBERSORT was applied for analysis [29]. The enrichment of immune-infiltrating cells and the expression of immune-related functions were analyzed via single-sample gene set enrichment analysis (ssGSEA) [30]. Statistical significance was defined as a *p* value and/or FDR ≤ 0.05.

### 2.4. Functional Analyses

DEGs were screened using the package of R language (|log_2_FC| > 0.585 and adj. *p* Val < 0.05). Gene Ontology (GO) analysis and Kyoto Encyclopedia of Genes and Genomes (KEGG) analyses were used to analyze the enriched pathways [31]. GO analysis determined biological processes, cellular components, and molecular function. Protein–Protein Interaction (PPI) network analysis was subsequently utilized to implement hub gene analysis according to the number of nodes [32].

### 2.5. Establishment and Validation of the Risk Model

The size of prognostic genes previously filtrated was narrowed down using univariate Cox regression analysis [33] and least absolute shrinkage and selection operator (LASSO) analysis [34]. The minimum lambda was regarded as the optimal value. Multivariate Cox regression [35] analysis determined several significant genes in establishing a risk model. The risk score was calculated using the following formula: Risk score = ΣExpn × βn, where Expn represents the expression value of each gene and βn represents the coefficient of the gene. Next, groups with high and low risk were separated. Survival analysis was performed using the Kaplan–Meier approach, and the predictive validity of the risk model was assessed using the receiver operating characteristic (ROC) [34]. A nomogram was constructed according to the status, age, FIGO stage, risk, and histopathological grade of CC patients.

## 3. Results

### 3.1. Identification of the Two Subtypes with Different OS

In total, 246 glycosylation-related genes were acquired. A total of 146 and 145 clinical samples were randomly classified into the training and testing cohorts, respectively. The Consensus Cluster Plus R package was used to cluster the CC patients in the training cohort. At K = 2, the optimal cluster stability was determined (Figure 1A–C). In total, clusters 1 and 2 each contained 180 patients and 126 patients, respectively. Cluster 1 showed better OS (*p* = 0.0003234; Figure 1D). These results indicated that CC patients could be classified into two subtypes with different OS.

### 3.2. Glycosylation of Proteins Can Affect Immune Function in the Two Molecular Subtypes

KEGG enrichment analysis showed that more pathways related to the glycosylation process were found in cluster 2 compared to cluster 1; accordingly, more immune-related pathways were found in subgroup 1 (Figure 2A). Thus, genes involved in glycosylation modification also act in the immune system. To investigate the association of glycosylation with the immune status, immune analyses were performed to explore differences in immunity between the two subgroups. According to the results of the ESTIMATE algorithm, CC patients in cluster 1 had significantly higher immune and ESTIMATE scores, and no appreciable differences were found in the stromal scores of patients in the two clusters (Figure 2B). In addition, the numbers of CD8 T cells, activated memory CD4 T cells, monocytes, M1 macrophages, resting dendritic cells, and resting mast cells were significantly higher in cluster 1 than in cluster 2, which was reversed as resting memory CD4 T cells, M0 macrophages, and activated mast cells showed no statistical significance for other immune-filtrating cells (Figure 2C).

Moreover, ssGSEA analysis illustrated that immune levels differed prominently between the two clusters, with cluster 1 having a comparatively high immune status, except for T helper 2 cells, while others were significantly higher in cluster 1 (Figure 2D). Moreover, cluster 1 showed significantly higher scores of immune activation and immunosuppression than cluster 2, except for the type II IFN response (Figure 2E). Cluster 1 had a higher immune status.

### 3.3. DEGs and Functional Analyses

In order to better investigate the underlying signaling pathways, functional analyses were performed, and 1195 DEGs were discovered in total. The result of GO analysis showed that the DEGs were more enriched in glycosylation, CC development, and biological processes related to the immune system, including the regulation of peptidase and endopeptidase activity, epidermal cell and keratinocyte differentiation, and the humoral immune response (Figure 3A,B). Meanwhile, some related cellular components and molecular functions were also enriched (Figure 3A). Moreover, several signaling pathways, including proteoglycans in cancer, cytokine–cytokine receptor interaction, the PI3K/AKT signaling pathway, the Hippo signaling pathway, and HPV infection were identified to be associated with glycosylation, the immune response, and CC (Figure 3C). PPI analysis of DEGs indicated that, compared with cluster 1, 565 genes were upregulated and 630 genes in cluster 2 were downregulated (Figure 3E). We selected the top 30 DEGs based on the number of nodes, including ITFB1 and SDC1, which were closely associated with the proliferation, migration, and prognosis of CC [36,37] (Figure 3D,F).

### 3.4. Risk Model Was Established in the Training Cohort

To establish the predictive model based on glycosylation-related genes in CC, we conducted univariate Cox regression analysis. Potential genes were screened using LASSO analysis, and 11 genes were selected with the optimal λ value (Figure 4A,B). Multivariate Cox analysis identified nine genes based on the genes generated through LASSO analysis to establish the risk model. The risk score = (0.0500580626412369 × MGAT4B) + (0.125800669064781 × FUT11) + (0.0302396698484961 × GALNT2) + (0.266509552440604 × DPY19L4) + (−0.152265202480328 × PMM1) + (0.0625200342212958 × GALNT10) + (−0.329088155870026 × MAN1C1) + (0.130257296074648 × COG3) + (−0.41914468450172 × DERL3). The risk model effectively classified CC patients into high- and low-risk groups (Figure 4C,D). In the high-risk group, the heatmap revealed that six candidate genes had a more general expression, except for PMM1, MAN1C1, and DERL3, and had a worse OS (Figure 4E,G). As for the model diagnosis for the risk model, for 1, 3, and 5 years, the area under the curve (AUC) of the ROC curve was 0.872, 0.865, and 0.841, respectively. The risk model had accurate 1-year predictive capability (Figure 4F). Finally, the TIME in both groups was assessed and the low-risk group had considerably higher stromal sores, immune scores, and ESTIMATE scores (Figure 4H).]

### 3.5. Risk Model Was Validated in the Testing and Total Cohorts

The testing cohort was separated into the high- and low-risk groups, and we validated the model in the two groups (Figure 5A,B). The expression of nine candidate genes was displayed on a heatmap with the same outcomes as the training cohort (Figure 5C). For 1, 3, and 5 years, the area under the curve (AUC) of the ROC curve was 0.558, 0.705, and 0.819, respectively (Figure 5D). The model demonstrated accurate 5-year predictive capability. According to the survival analysis, the high-risk group had considerably worse OS in the testing cohort (*p* = 0.011; Figure 5E). Similar to the training cohort, the low-risk group had significantly higher immune scores and ESTIMATE scores (Figure 5F). In addition, all 291 samples in the total cohort were analyzed to validate the constructed risk model (Figure 6). The results were similar to those for the testing cohort.

The risk model and clinical data including the age, FIGO stage, risk, and histopathological grade of CC patients were incorporated into a nomogram to more accurately predict the prognosis of CC patients (Figure 7A). The risk score and clinical risk factors were endowed with a certain score according to their impact on the prognosis in CC. The C-index of the nomogram reached 0.736 (se = 0.04). We then validated the nomogram in all samples. As for the diagnosis of the nomogram, the calibration curve showed consistency between the expected and observed OS (Figure 7B).

## 4. Discussion

Protein glycosylation refers to the enzymatic attachment of a glycosyl donor to the side groups of amino acids [38]. It is one of the most abundant post-translational modifications in eukaryotic cells, which is also a critical process involved in numerous gynecological malignancies, including CC [39]. Glycosylation influences extensive aspects of the CC biology, including cell–cell adhesion, cell surface expression, and cancer signaling.

CC cells can combine different O-glycosylation modifications and alter the expression levels of proteins to govern their malignant phenotypes [40]. The Tn antigen refers to GalNAc-Ser/Thr during the biosynthesis of mucin-type O-glycosylation. Mutations in the Tn antigen exert a significant impact on tumor cell adhesion, immune evasion, and migration [41]. The abnormal glycosylation profile and Tn-antigen-induced cell identification both contribute to the pathogenesis of CC [42]. Glycosyltransferases are involved in most glycans’ biosynthesis. The altered expression of glycosyltransferases in CC leads to more aggressive characteristics and drug resistance [43].

Under normal conditions, cellular immunity is regulated by activation signals (co-stimulatory molecules) and inhibition signals (immune checkpoints) [44] to maintain homeostasis. An immunosuppressive tumor microenvironment is produced as CC cells continue to evade immune surveillance. It includes the upregulation of regulatory T cells (Tregs) while downregulating anti-cancer activity by effector T cells, the loss of major histocompatibility complex antigen presentation, and the upregulation of immune checkpoints [45].

The programmed cell death receptor (PD-1) on the external effector immune cells binds the programmed cell death receptor ligand (PD-L1) produced by cancer cells. The PD-1/PD-L1 axis is a major immune checkpoint mechanism [46]. According to a study, cisplatin-based treatment can increase PD-L1 in CC, and utilizing a checkpoint blocker may help with tumor cell regression [47]. Pembrolizumab, a PD-1 inhibitory antibody, has been authorized for persistent, recurrent, or metastatic CC treatment [48]. In addition, glycosylation also plays an important role in this pathway. Four glycosylation sites on PD-L1 in the extracellular domain serve as the primary sites of N-glycan modification [49]. Via EGFR signaling and EMT, 1,3-N-acetylglucosamine transferase 3 catalyzes the increase in interaction with PD-1 in triple-negative breast cancer [50,51]. Many cancer types, including melanoma, cervical cancer, and non-small-cell lung cancer, have been discovered to exhibit PD-L1 glycosylation, which is a typical characteristic of cancer [52]. Moreover, the extracellular domain of PD-1 also has four N-glycosylation sites, and glycosylation is necessary to preserve the stability of the PD-1 protein and the location of the cell surface [53]. In T cells, PD-1 is extensively N-glycosylated, and its particular type varies when the TCR is activated [54]. According to a previous study, camdelizumab, a PD-1 antibody, can specifically bind N58 glycosylated PD-1 and block the PD-1/PD-L1 pathway [55]. Although checkpoint inhibitors have achieved extraordinary progress in cancer [56,57], they have attracted researchers’ attention to the exploration of predictive biomarkers for CC owing to their limited efficacy.

In our study, two molecular subgroups were identified through consensus clustering according to the glycosylation-related gene expression matrix of CC patients. Immune analysis showed that cluster 1 had a higher immune status, and poor prognosis among CC patients in the high-risk group was found to be related to the immunosuppressive tumor microenvironment. Then, the results of the function assay confirmed that the expression of DEGs was associated with immune dysregulation and glycosylation. We carried out univariate and multivariate Cox regression and LASSO analyses to investigate the clinical value of these genes in CC further, and nine prognostic genes (MGAT4B, FUT11, GALNT2, DPY19L4, PMM1, GALNT10, MAN1C1, COG3, and DERL3) were finally identified to establish the risk model. High- and low-risk groups of CC patients were identified according to the model. To check the efficacy of the constructed model, we further validated these nine genes and our prognostic model in the testing and the total cohorts. The AUC values of ROC for 5 years were 0.841, 0.819, and 0.815, respectively, in the training, testing, and total cohorts, indicating that the constructed model was accurate in predicting the prognosis. Survival analysis revealed that regardless of the cohort, the constructed risk model demonstrated a robust predictive level for the survival of CC patients. Significant decreases in the ESTIMATE score and immune score were always accompanied by poor survival. Therefore, the constructed risk model was proven to be significantly connected with the TIME and possessed strong potential to forecast the prognosis of CC patients in the training, testing, and total cohorts. Finally, a nomogram combining clinical features and the risk score was also created and calibrated, which demonstrated excellent potential in forecasting CC survival. The above results confirmed that the developed risk model could accurately predict the prognosis in CC; TIME disorders, including lower immune and ESTIMATE scores, were prevalent in individuals with poor prognosis.

Fucosyltransferase 11 (FUT11) has been reported as a new biomarker for CC prognosis [58]. Mucin O-glycosylating enzyme (GALNT2) has been reported to exhibit the capacity to serve as a novel biomarker for endometrial hyperplasia [59]. GALNT10 was found to be highly predictive of the OS of ovarian cancer [60]. Increased GALNT10 expression also promotes tumor growth by creating an immunosuppressive microenvironment and is associated with poor clinical outcomes in those with high-grade ovarian serous carcinoma [61].

Integrinβ1 (ITGB1) is markedly overexpressed in various malignancies and has been reported as a prospective marker in predicting the effects of immunotherapy in gastric cancer [62]. A previous study indicated that Syndecan 1 (SDC1) might be a novel immune-related prognostic biomarker for pancreatic adenocarcinoma [63]. However, our study is the first to illustrate that these glycosylation-related genes may have a functional involvement in CC and are related to immune infiltration.

Furthermore, KEGG analysis revealed that the DEGs were prominent in the PI3K/AKT signaling pathway, cytokine–cytokine receptor interaction, and Hippo signaling pathway. By influencing cell survival, proliferation, and migration, the aberrant triggering of the PI3K/AKT/mTOR signaling pathway can result in a malignant phenotype of cancer cells and chemotherapy resistance [64]. It is crucial for the crosstalk between the virus and the host cells in HPV-positive cancer cells. E6 and E7 have been found to have an activating effect on AKT and mTOR [65,66]. Normally, cytokines are secreted glycoproteins that promote cellular proliferation, differentiation, and apoptosis [67]. However, immuno-suppressive cells are recruited when cytokines bind to their cognate receptors, resulting in tumor invasion and metastasis [68]. The nuclear accumulation of downstream effector factor YAP of the Hippo pathway stimulates the expression of EGF-like ligands (such as TGF-α), which activates EGFR, thereby promoting the growth and invasion of CC. The proteasome-dependent YAP protein can be prevented from degradation and maintained at high levels in CC cells by HPV E6/E7 oncoproteins [69]. The Hippo pathway is significant to cell–cell junctions, as well as [70] extracellular matrix attachment [71] and the TIME [72].

Our predictions were based on analyses of online databases, which is one limitation of our study. As a result, additional experimental validation is required.

## 5. Conclusions

In this study, we discovered that the expression of glycosylation-related genes was highly correlated with the TIME and enriched in several significant pathways in CC. Our research might offer a new target for the prognosis of CC. Additional research on these genes and associated signaling networks may reveal new perspectives on CC immunotherapy and lead to better prognoses.

## Figures and Tables

**Figure 1 jpm-13-00529-f001:**
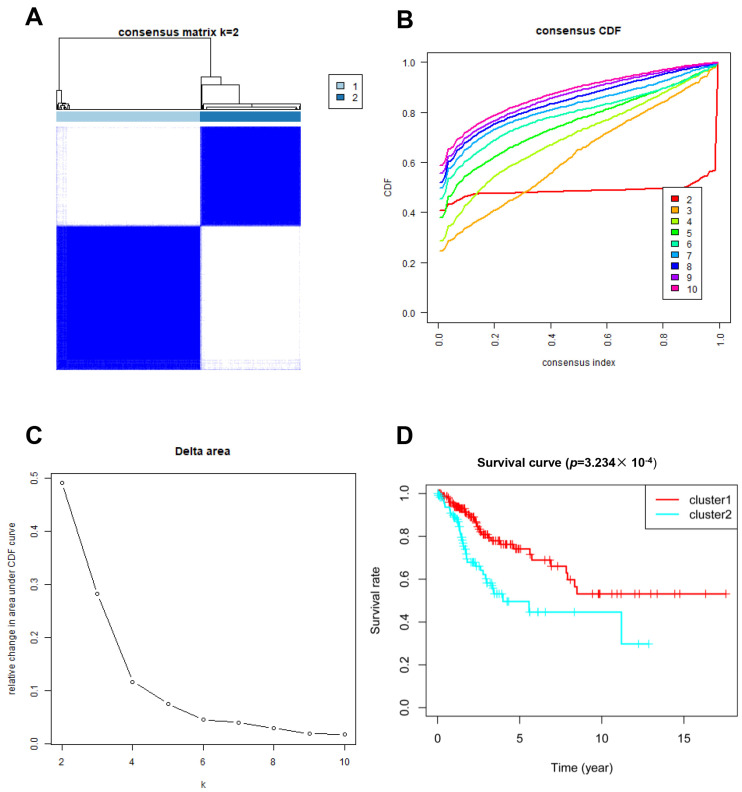
Clustering analysis based on the expression of glycosylation-related genes. (**A**–**C**) Consensus clustering for k = 2; (**D**) Patient survival curves for the two subgroups.

**Figure 2 jpm-13-00529-f002:**
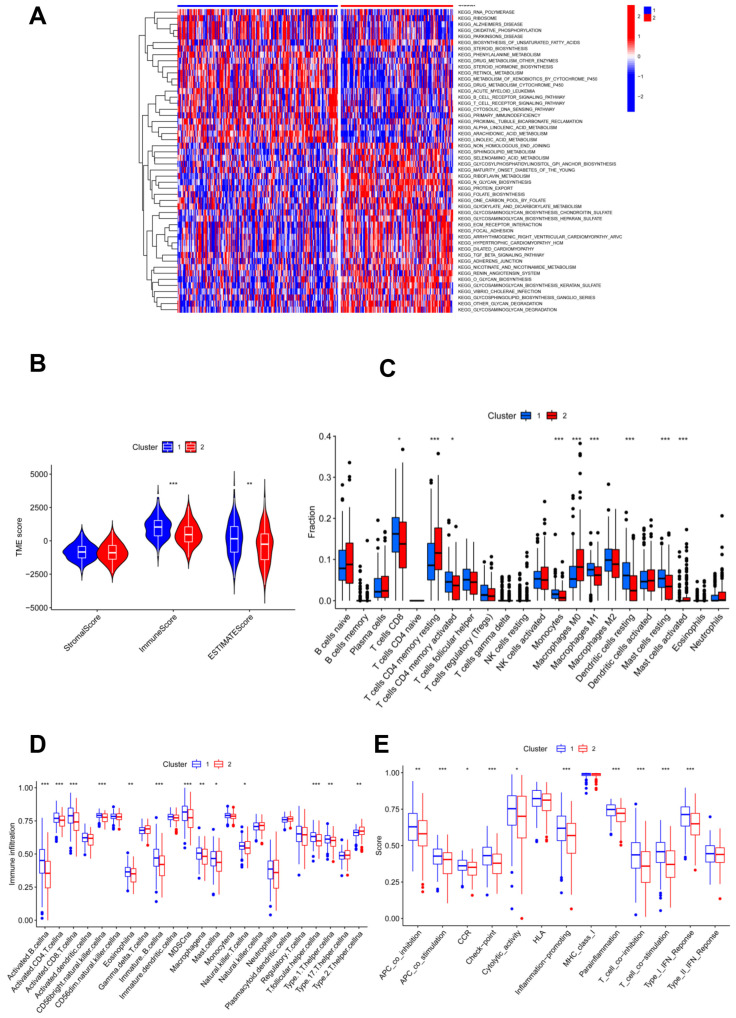
Immune characteristic analyses between two clustered subgroups. (**A**) Heatmap illustrating the results of signaling pathways enriched by KEGG analysis; (**B**) stromal score, immune score, and ESTIMATE score in two clustered subgroups; (**C**) fraction of 22 immune-filtrating cells evaluated by CIBERSORT; (**D**) the differential enrichment levels of 23 immune-related cells are displayed in boxplots; (**E**) ssGSEA algorithm analyses. * *p* < 0.05; ** *p* < 0.01; *** *p* < 0.001.

**Figure 3 jpm-13-00529-f003:**
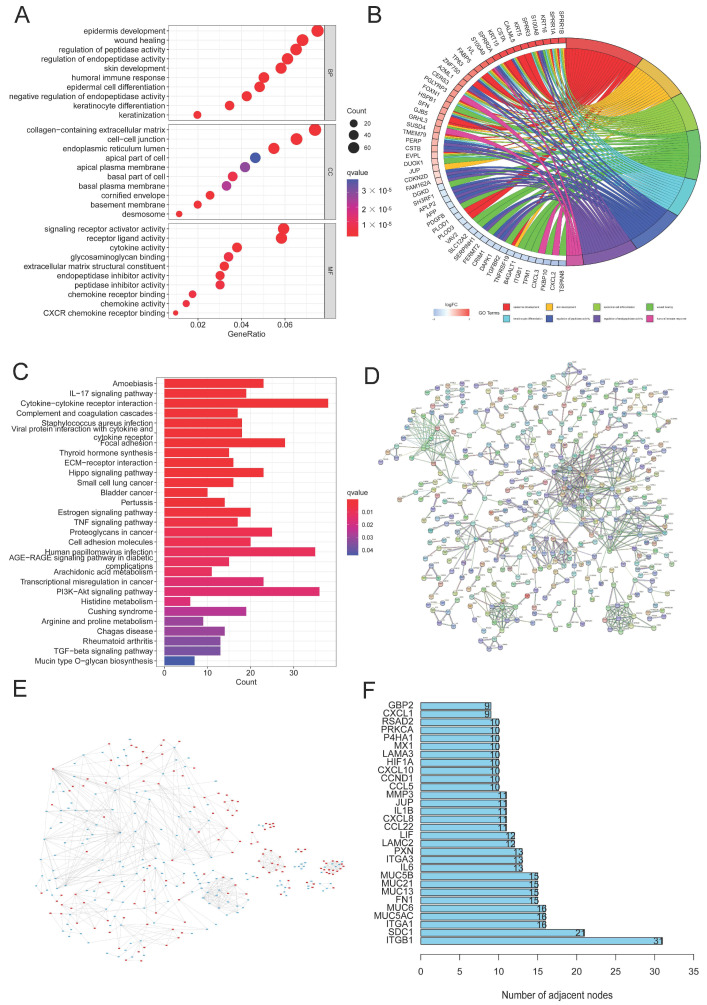
DEG and function assays. (**A**,**B**) Bubble diagram and circle plot visualizing the results of GO analysis; (**C**) KEGG analysis exploring the enriched signaling pathways; (**D**) PPI analysis of DEGs; (**E**) compared to cluster 1, the color of nodes reflects the upregulated genes (red) and downregulated genes (blue) in cluster 2; (**F**) the top 30 DEGs with the largest numbers of nodes were chosen through PPI analysis.

**Figure 4 jpm-13-00529-f004:**
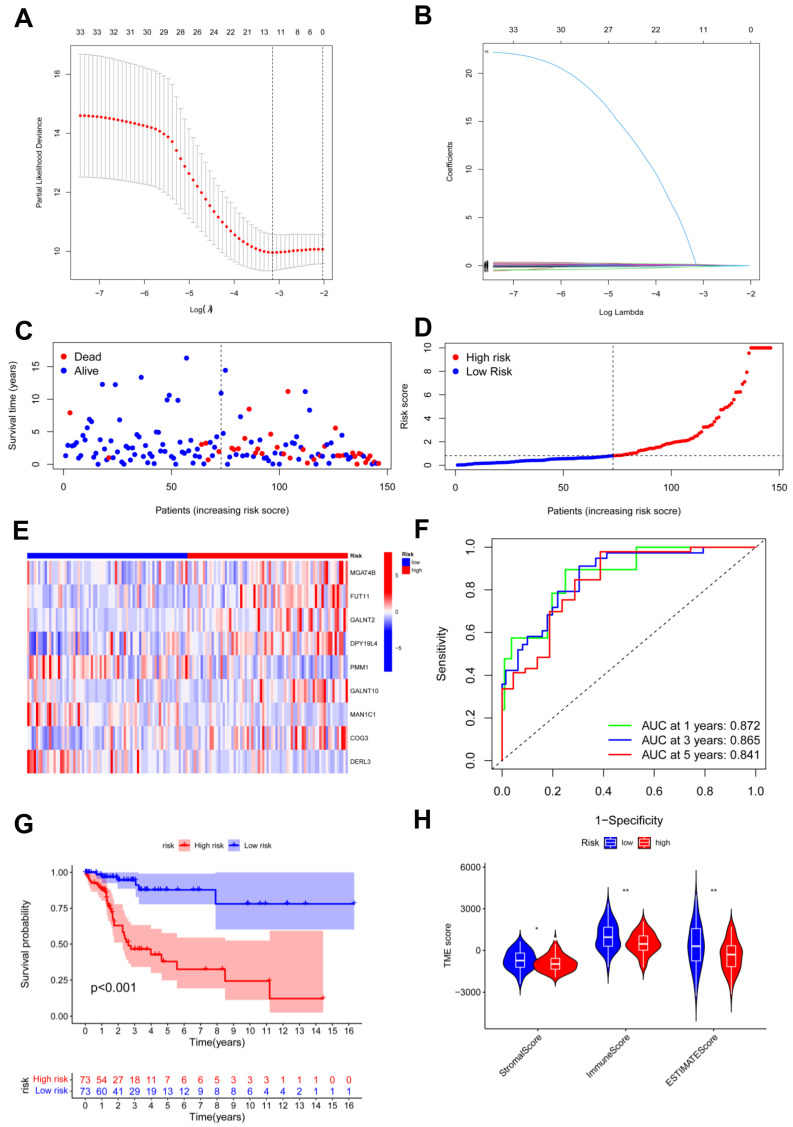
Identification and construction of risk model. (**A**,**B**) LASSO analysis with minimal lambda; (**C**,**D**) assessments of the survival status and risk scores in high- and low-risk groups; (**E**) the risk gene expression in the two groups; (**F**) ROC curve of the risk model; (**G**) survival curves in the two groups; (**H**) stromal, immune, and ESTIMATE scores of the two groups. * *p* < 0.05; ** *p* < 0.01.

**Figure 5 jpm-13-00529-f005:**
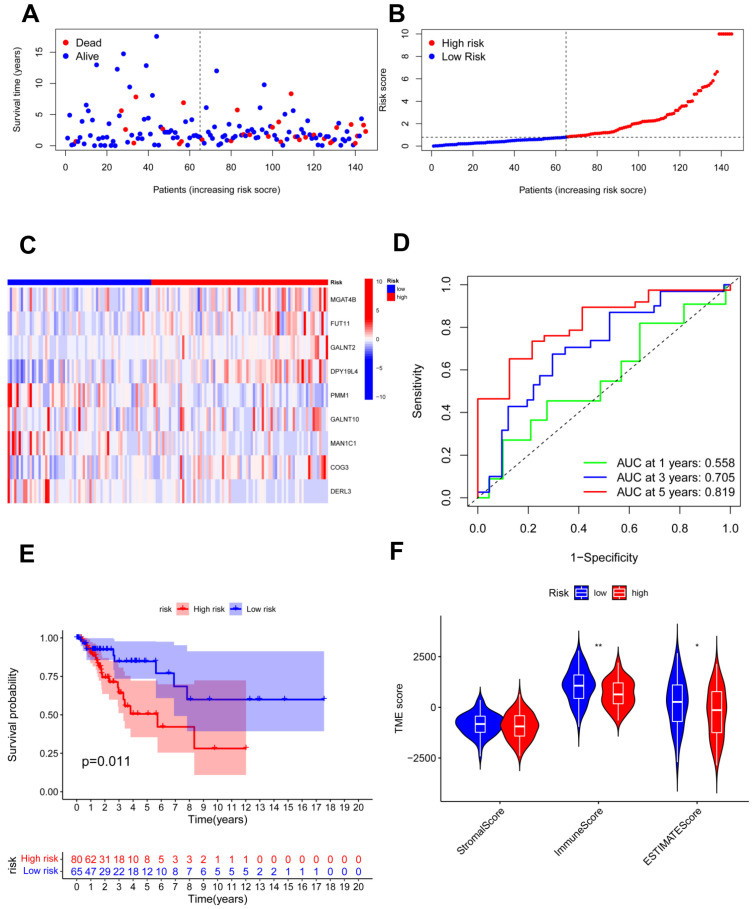
Validation of the risk model in the testing cohort. (**A**,**B**) Assessments of the survival status and risk scores in the two groups; (**C**) the risk gene expression in the testing cohort; (**D**) ROC curve of the risk model; (**E**) survival curves in the two groups; (**F**) stromal, immune, and ESTIMATE scores of the two groups. * *p* < 0.05; ** *p* < 0.01.

**Figure 6 jpm-13-00529-f006:**
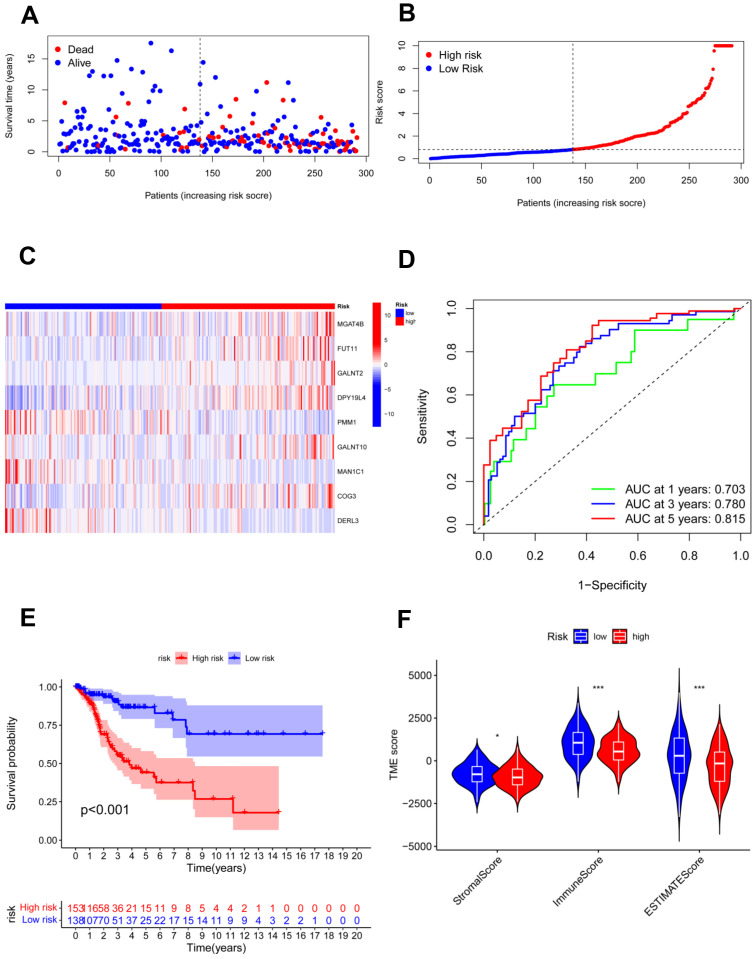
Validation of the risk model in the total cohort. (**A**,**B**) Assessments of the survival status and risk scores in the two groups; (**C**) the risk gene expression in the total cohort; (**D**) ROC curve in the total cohort; (**E**) survival curves of the CC patients in the two groups; (**F**) stromal, immune, and ESTIMATE scores of the two groups. * *p* < 0.05; *** *p* < 0.001.

**Figure 7 jpm-13-00529-f007:**
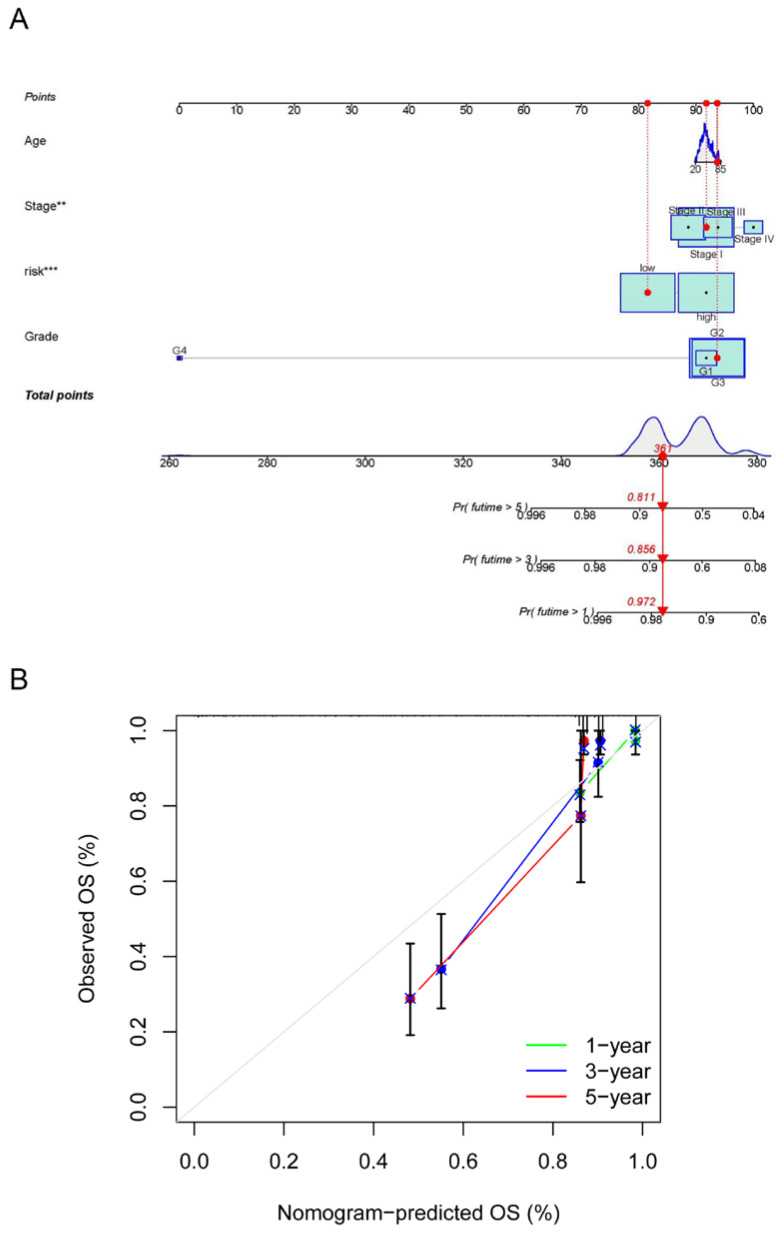
Establishment and calibration of a predictive nomogram. (**A**) The nomogram for predicting OS in CC patients; (**B**) calibration plots of the constructed nomogram in all samples. ** *p* < 0.01, *** *p* < 0.001.

**Table 1 jpm-13-00529-t001:** Clinical and Pathologic Characteristics of The Patient with Cervical Cancer.

Variable	Training Dataset					Validation Dataset				
	Total	Risk Group		χ2	*p* Value	Total	Risk Group		χ2	*p* Value
		Lower	Higher				Lower	Higher		
	n = 130	n = 70	n = 60			n = 128	57	71		
Age, y										
≤45	57	31	26	0.012	0.913	65	29	36	0.000	0.984
>45	73	39	34			63	28	35		
FIGO stage										
I	70	37	33	0.060	0.807	76	36	40	0.610	0.435
II–IV	60	33	27			52	21	31		
Grade										
G1	11	9	2	4.780 ^a^	0.189	7	3	4	0.171984 ^a^	0.918
G2	60	31	29			67	31	36		
G3	58	29	29			54	23	31		
G4	1	1	0			0	0	0		

^a^, Likelihood Ratio.

## Data Availability

The data presented in this study are available on request from the corresponding author.

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
