# Peer review of "Association of Glycosylation-Related Genes with Different Patterns of Immune Profiles and Prognosis in Cervical Cancer"

_jpm, 2023, doi:10.3390/jpm13030529_

Round 1

Reviewer 1 Report

WHICH TYPE OF GRADIND AND STAGE  ..THESE DATA NEEDS TO CLEARLY  EXSPLAINED 

Author Response

Question 1:WHICH TYPE OF GRADIND AND STAGE THESE DATA NEEDS TO CLEARLY EXSPLAINED.

Response: We feel sorry that we did not provide enough information and caused you confusion. The grade represents the histopathological grade. The higher the grade, the higher the malignant degree of cervical cancer. The stage represents FIGO stage. We have presented the patient information related to clinical grading and staging in the form of table. Relevant part was highlighted in the modified version on line 112-115, 148-149,220-222 and Table 1.

Reviewer 2 Report

The manuscript by Dr. Jing et al. investigates the if glycosylation of proteins could have an impact on immune microenvironment regulation and prognosis in cervical cancer patients.

Nowadays the increase interest on biological and molecular pathways associated with the cervical cancer progression makes the present article really interesting. Anyway, some drawbacks are present:

Title:

- since the final purpose of the study is to build a predictive model for cervical cancer patients survival based on glycosylation-related genes, i would suggest the authors revise the title to highlight this goal

Introduction:

- the main objective of the study is not clearly specified. the authors state that it is "...to investigate how glycosylation affects the TIME and prognosis ...". However, the study classifies patients with cervical cancer into 2 groups according to their prognosis. Subsequent analysis compares the distribution of glycosylation pathways, TIME, and gene expression between these two groups (finding that the 2 groups differ in all of these categories). however, from this it can be deduced that the glycosylation pathways are correlated with the prognosis (a fact already known in the literature e.g.: Martinez-Morales P et al., PeerJ. 2021; Xu Zet al., Front Oncol. 2021; Shen H et al., Diagnostics. 2022) but does not indicate the mechanisms by which glycosylation would affect the TIME

Methods:

- please see the previous section

- in the Methods section there are some results (e.g. the number of genes selected). Please move these data in the “results” section.

- I suggest the authors enrich this section explaining why they chose to conduct those specific analyses

- Please provide the correct references of all the tests used

-

Results:

- In this section there are several comments (especially enthusiastic comments) regarding the results of the analysis conducted. Please consider that all comments (both positive and negative) and considerations on the interpretation of the results should be moved to the "Discussion" chapter

- It was found that the 2 groups of patients differ in the amount of pathways related to glycosylation, in the immune score and in the type of immune cell population. However, this does not directly demonstrate that glycosylation correlates directly with TIME

- Please provide information about clinical characteristics of the patients. It is really relevant also due to the fact that some of these are included in the final nomogram.

Manuscript require minor English revision.

Author Response

Comments and Suggestions for Authors

The manuscript by Dr. Jing et al. investigates the if glycosylation of proteins could have an impact on immune microenvironment regulation and prognosis in cervical cancer patients.

Nowadays the increase interest on biological and molecular pathways associated with the cervical cancer progression makes the present article really interesting. Anyway, some drawbacks are present:

Question 1: Title: since the final purpose of the study is to build a predictive model for cervical cancer patients survival based on glycosylation-related genes, i would suggest the authors revise the title to highlight this goal

Response: We feel great thanks for your professional review work on our article. We have carefully considered your suggestion and made modifications. The current topic is “Expression of Glycosylation-Related Genes Is Involved in the Immune Microenvironment and Predicts Prognosis in Cervical Cancer”.

Question 2:Introduction: the main objective of the study is not clearly specified. the authors state that it is "...to investigate how glycosylation affects the TIME and prognosis ...". However, the study classifies patients with cervical cancer into 2 groups according to their prognosis. Subsequent analysis compares the distribution of glycosylation pathways, TIME, and gene expression between these two groups (finding that the 2 groups differ in all of these categories). however, from this it can be deduced that the glycosylation pathways are correlated with the prognosis (a fact already known in the literature e.g.: Martinez-Morales P et al., PeerJ. 2021; Xu Zet al., Front Oncol. 2021; Shen H et al., Diagnostics. 2022) but does not indicate the mechanisms by which glycosylation would affect the TIME.

Response: Thank you for your comments. One of the primary characteristics of tumor cells is the glycosylation of the glycoproteins and glycolipids found on the cell surface. Tumor cells express glycosylation differently from the way in which normal cells do. As a result of the large number of different types of glycosylation-dependent lectin receptors expressed by immune cells, these cells are able to detect changes in glycosylation in the environment, which may induce immunosuppression [14]. Tumor cells can also camouflage themselves by expressing host-derived glycosylation and affect the expression of antigen-presenting cells, M2 macrophages, T cells, and NK cells can promote immune escape [15]. Carbohydrate Lewis antigens can attach to carcinoembryonic antigens expressed in colon cancer cells and combine with C-type lectin expressed in macrophages and immature dendritic cells to induce innate immune suppression [16, 17]. The aggregation of Treg cells, the minimal infiltration of effector T cells, and the activity of NK cells are all strongly correlated with the sialylated structure of melanoma cells and the progression of the tumor in vivo [18]. The glycosylation of tumor cells usually occurs in the early stage of tumor development. In the prophase lesions of different types of tumors, some tumor-related glycosylation expression has appeared [19].  We have supplemented the above mechanism of glycosylation affecting tumor immune microenvironment in the modified version (line 70-85).

Question 3: Methods: please see the previous section

in the Methods section there are some results (e.g. the number of genes selected). Please move these data in the “results” section.

Response: We gratefully appreciate for your valuable suggestion. We have moved some data in the methods part to the result section. We have amended the statement as follows (line 127,143, 152-153).

Question 4: I suggest the authors enrich this section explaining why they chose to conduct those specific analyses

Response: Thank you for your comments. In our manuscript, according to your valuable suggestions, relevant part was highlighted in the modified version on line 117,120-122.

Question 5: Please provide the correct references of all the tests used

Response: Thanks for your reminder. We have added the correct reference of all analyses in appropriate section (line 121-148).

Question 6:

Results:

In this section there are several comments (especially enthusiastic comments) regarding the results of the analysis conducted. Please consider that all comments (both positive and negative) and considerations on the interpretation of the results should be moved to the "Discussion" chapter.

Response: Thank you for your comments. We found that these comments were inappropriate in results section, so we deleted them, and we also showed relevant comments in the discussion chapter. relevant part was highlighted in the modified version on line 269-273, 284-286, 276-277.

Question 7: It was found that the 2 groups of patients differ in the amount of pathways related to glycosylation, in the immune score and in the type of immune cell population. However, this does not directly demonstrate that glycosylation correlates directly with TIME.

Response: We feel great thanks for your professional review work on our article. Our previous results indicated glycosylation may participate in the process of immune microenvironment in cervical cancer. This is a suggestive result, and we will further explore the relationship between them in depth though experiments.

Question 8: Please provide information about clinical characteristics of the patients. It is really relevant also due to the fact that some of these are included in the final nomogram.

Response: We have supplemented information about clinical characteristics of the patients in Table 1.

Question 9: Manuscript require minor English revision.

Response: We appreciate the comments. We would use the recommended editing services to polish the English of the manuscript.

Reviewer 3 Report

The authors provide the new pathway for clinical target for Cervical Cancer using available data base. They are trying to explore the relationship among glycosylation-related genes, immune microenvironment, and prognosis of cervical cancer which can use for further study by other researcher for good clinical outcome. These research are offering a new target for the prognosis of Cervical Cancer and new perspectives on CC immunotherapy and lead to better prognoses which is very necessary to the health related to malignancy diseases.

This manuscript is for the most part well written with substantial discussion of results and postulated according to the evidence provided. The references are appropriate and timely.

Minor criticisms

• Please undergo a thorough check of the manuscript for typographical and grammatical errors.

Author Response

The authors provide the new pathway for clinical target for Cervical Cancer using available data base. They are trying to explore the relationship among glycosylation-related genes, immune microenvironment, and prognosis of cervical cancer which can use for further study by other researcher for good clinical outcome. These research are offering a new target for the prognosis of Cervical Cancer and new perspectives on CC immunotherapy and lead to better prognoses which is very necessary to the health related to malignancy diseases.

This manuscript is for the most part well written with substantial discussion of results and postulated according to the evidence provided. The references are appropriate and timely.

Minor criticisms

Question: Please undergo a thorough check of the manuscript for typographical and grammatical errors

Response: We appreciate the comments. We would use the recommended editing services to polish the English of the manuscript.

Round 2

Reviewer 2 Report

I appreciate the effort made by the authors to improve their manuscript. however, the methodology and the analyzes conducted do not allow to define the correlation and relationship between Glycosylation-Related Genes and Immune Microenvironment in cervical cancer patients. I would suggest the authors set their paper on what appears to be their primary goal: build a nomogram that, including a "molecular" risk score, could predict cervical cancer patients' survival.
